# A New Cell Line from the Brain of Red Hybrid Tilapia (*Oreochromis* spp.) for Tilapia Lake Virus Propagation

**DOI:** 10.3390/ani14111522

**Published:** 2024-05-22

**Authors:** Aslah Mohamad, Matepiya Khemthong, Pirada Trongwongsa, Tuchakorn Lertwanakarn, Piyathip Setthawong, Win Surachetpong

**Affiliations:** 1Department of Veterinary Microbiology and Immunology, Faculty of Veterinary Medicine, Kasetsart University, Bangkok 10900, Thailand; aslahumt@gmail.com (A.M.); matepiya@hotmail.com (M.K.); piradatr@gmail.com (P.T.); 2Department of Physiology, Faculty of Veterinary Medicine, Kasetsart University, Bangkok 10900, Thailand; tuchakorn.l@ku.th; 3Laboratory of Biotechnology, Chulabhorn Research Institute, Bangkok 10210, Thailand

**Keywords:** tilapia lake virus, red hybrid tilapia, cell line, virus propagation, culture conditions

## Abstract

**Simple Summary:**

Simple Summary: In this study, a new cell line derived from red hybrid tilapia brain tissue, RHTiB, was established. This fibroblast-like cell line was maintained for over 50 passages with optimal growth at 25 °C in Leibovitz-15 medium with 10% fetal bovine serum at pH 7.4. Genetic and chromosomal analyses confirmed its origin from red hybrid tilapia. Additionally, immunofluorescence and RT-qPCR tests revealed successful TiLV propagation in RHTiB cell lines. The development of this novel cell line offers valuable prospects in enhancing diagnostic techniques in red hybrid tilapia.

**Abstract:**

Tilapia lake virus (TiLV) presents a substantial threat to global tilapia production. Despite the development of numerous cell lines for TiLV isolation and propagation, none have been specifically derived from red hybrid tilapia (*Oreochromis* spp.). In this study, we successfully established a new cell line, RHTiB, from the red hybrid tilapia brain. RHTiB cells were cultured for 1.5 years through over 50 passages and demonstrated optimal growth at 25 °C in Leibovitz-15 medium supplemented with 10% fetal bovine serum at pH 7.4. Morphologically, RHTiB cells displayed a fibroblast-like appearance, and cytochrome oxidase I gene sequencing confirmed their origin from *Oreochromis* spp. *Mycoplasma* contamination testing yielded negative results. The revival rate of the cells post-cryopreservation was observed to be between 75 and 80% after 30 days. Chromosomal analysis at the 25th passage revealed a diploid count of 22 pairs (2n = 44). While no visible cytopathic effects were observed, both immunofluorescence microscopy and RT-qPCR analysis demonstrated successful TiLV propagation in the RHTiB cell line, with a maximum TiLV concentration of 10^7.82 ± 0.22^ viral copies/400 ng cDNA after 9 days of incubation. The establishment of this species-specific cell line represents a valuable advancement in the diagnostic and isolation tools for viral diseases potentially impacting red hybrid tilapia.

## 1. Introduction

Tilapia (*Oreochromis* spp.) is known for its fast growth, adaptability to different environments, and resistance to various microbial diseases [1,2] and is thus a commercially important fish species in aquaculture. The global production of tilapia has expanded in Asia, Africa, and the Middle East, with the Food and Agriculture Organization (FAO) reporting record production exceeding 6 million tons in 2020, which makes tilapia the second most farmed fish species after carp [3]. However, the emergence and rapid spread of the highly virulent tilapia lake virus (TiLV), officially named by the International Committee on Taxonomy of Viruses as *Tilapinevirus tilapiae* [4], poses a significant threat to the worldwide tilapia industry and has resulted in considerable economic losses [5,6,7,8].

Since its initial identification in Israel in 2014 [5], TiLV has been detected in more than 17 countries [9,10], with recent outbreaks in Vietnam [11] and China [12]. TiLV infections have been associated with high mortality rates of up to 90% in red tilapia (*Oreochromis* spp.) [9]. Such infections are characterized by severe clinical signs, such as lethargy, ocular abnormalities, and skin erosion [5,9]. While a variety of diagnostic methods have been developed to detect TiLV, including molecular assays [13], cell culture remains the gold standard for virus isolation [14,15].

Despite the high susceptibility of various cell lines derived from different piscine species to TiLV infection, such as those from largemouth bass (*Micropterus salmoides*), hybrid snakehead (*Channa argus* × *Channa maculata*), and perch (*Siniperca chuatsi*) [16], the host specificity of viruses requires the use of appropriate cell lines for viral isolation and propagation [17]. While TiLV may infect other fish cells, such as E-11 cells, which are commonly used nowadays for TiLV propagation and isolation [5], previous studies suggest that TiLV is highly host-specific, mainly infecting tilapia cell lines [16,18,19,20,21,22]. However, no primary cell line derived from red hybrid tilapia has been established or tested for TiLV propagation. In this study, we therefore aimed to establish a new cell line from red hybrid tilapia, to determine its optimal growth conditions, and to evaluate its susceptibility to TiLV propagation.

## 2. Materials and Methods

### 2.1. Tissue Preparation for Primary Cell Cultures

Red hybrid tilapia fingerlings aged 2 months and weighing 10.58 ± 3.71 g were obtained from a hatchery farm in Phetchaburi Province, Thailand. The fish were euthanized using eugenol at a concentration of 2 mL per 1000 mL of water, in line with the guidelines approved by the Institutional Animal Care and Use Committee of Kasetsart University (protocol number ACKU64-VET-071) for animal handling and care.

The primary cell culture was attempted by following the tissue explant method [23]. In brief, the organs (brain, liver, spleen, heart, gills, fins, and ocular muscles) were aseptically dissected, washed three times with sterile phosphate-buffered saline (PBS) supplemented with an antibiotic–antimycotic solution (200 U/mL of penicillin, 0.2 mg/mL of streptomycin, and 0.5 µg/mL of Amphotericin B; Sigma-Aldrich, St Louis, MO, USA), and cut into 1 mm^3^ pieces using sterile scissors. The tissues were then seeded in triplicate into 25 cm^2^ cell culture flasks (Corning Glass Work, Corning, NY, USA) containing 5 mL of Leibovitz’s L-15 medium at pH 7.4 supplemented with 20% fetal bovine serum (FBS; Gibco, Buffalo, NY, USA) and the antibiotic–antimycotic solution in a low-temperature incubator without CO_2_ at 25 °C. The medium was replaced every 7 days, and the cultures were monitored daily using an inverted microscope (CKX53, Olympus, Tokyo, Japan) at 40× magnification.

### 2.2. Subculture and Maintenance

Upon reaching 80–90% confluence, the cells were washed twice with sterile PBS (pH 7.4) and trypsinized with 0.125% trypsin–ethylenediaminetetraacetic acid (Thermo Fisher Scientific, Grand Island, NY, USA). The detached cells were suspended in fresh L-15 medium supplemented with 20% FBS, 200 U/mL of penicillin, 0.2 mg/mL of streptomycin, and 0.5 µg/mL of Amphotericin B and centrifuged at 500 RCF at 4 °C for 5 min. The supernatant was then discarded. The cell pellet was resuspended in L-15 medium containing 20% FBS and seeded into new 25 cm^2^ cell culture flasks with L-15 medium and 20% FBS. Subsequent passages were split at a 1:2 ratio from the second passage onward. After the 10th passage, the cells were maintained in L-15 medium containing 10% FBS, 100 U/mL of penicillin, 0.1 mg/mL of streptomycin, and 0.25 µg/mL of Amphotericin B. The medium was replaced every 7 days, with daily monitoring under an inverted microscope (CKX53, Olympus, Tokyo, Japan) at 40× magnification.

### 2.3. Cryopreservation and Recovery

The cells at 75–90% confluence were collected via trypsinization for cryopreservation. Following centrifugation at 500 RCF at 4 °C for 5 min, the supernatant was discarded, and the cell pellet was resuspended in a freezing medium consisting of 10% dimethyl sulfoxide in 90% FBS. The suspensions were aliquoted into sterile cryovials and stored overnight at −80 °C before being transferred to liquid nitrogen for long-term storage. After 30 days, the frozen cells were thawed in a water bath at 37 °C. The frozen medium was removed, and the cells were suspended in L-15 medium with 10% FBS, 100 U/mL of penicillin, 0.1 mg/mL of streptomycin, and 0.25 µg/mL of Amphotericin B. The cell viability was assessed using trypan blue staining, and the cell count was obtained with a hemocytometer.

### 2.4. Cell Growth Characteristics

To determine the optimal growth conditions, including the FBS concentrations and pH levels, the cells at passage 20 were seeded in duplicate at a density of 9.0 × 10^3^ cells/well in 96-well cell culture plates. For FBS concentration optimization, the cells were seeded separately in L-15 media containing 5%, 10%, 15%, or 20% FBS. For pH optimization, the cells were seeded in 100 µL of L-15 medium with 20% FBS and pH levels ranging from pH 7 to 8. The pH of the L-15 medium was adjusted by adding 1 N hydrochloric acid or 1 N sodium hydroxide. The cell viability was quantified on days 1, 3, 5, and 7 using a Cell Counting Assay Kit-8 (Sigma-Aldrich, St. Louis, MO, USA), in line with a previously described procedure [24]. The optimal temperature for cell growth was assessed by incubating the cells in duplicate in 24-well cell culture plates (6.0 × 10^4^ cells/well) at 25 °C, 28 °C, and 30 °C. The cell viability was determined on days 1, 3, 5, and 7 by trypan blue staining and quantified using the Countess™ II Automated Cell Counter (Invitrogen, ThermoFisher Scientific, Waltham, MA, USA).

### 2.5. Identification of Origin Cell Line Using Mitochondrial Markers

To verify the specificity of the new cell line, which we named red hybrid tilapia brain (RHTiB) cells, the amplification of the mitochondrial cytochrome oxidase subunit 1 (*cox1*) gene was conducted. The genomic DNA was isolated from the RHTiB cells using a Quick-DNA Miniprep Plus Kit (Zymo Research, Orange, CA, USA), in accordance with the manufacturer’s instructions. The fragments of the *cox1* gene were amplified with the Fish F1 and Fish R1 universal primers under polymerase chain reaction (PCR) conditions, as described by Ward et al. [25]. The amplified product was visualized in 1.5% agarose gel containing ethidium bromide and subjected to Sanger sequencing (1st Base, Singapore). The forward and reverse nucleotide sequences of the *cox1* gene were assembled using CLC Genomic Workbench version 11.0.1 (Qiagen, Hilden, Germany), and the *cox1* nucleotide sequence was deposited in the National Center for Biotechnology Information GenBank (www.ncbi.nlm.nih.gov/genbank, accessed on 15 January 2023). A maximum likelihood phylogenetic tree and a genetic distances table were subsequently generated for different fish species from the *cox1* nucleotide sequence using the Kimura 2-parameter model in the Molecular Evolutionary Genetics Analysis X (MEGAX) tool [26].

### 2.6. Detection of Mycoplasma and Snakehead Retrovirus Contamination in RHTiB Cell Line

We screened for *Mycoplasma* contamination in the RHTiB cells after passages 10 and 20 using the LookOut Mycoplasma PCR Detection Kit (Sigma-Aldrich, St Louis, MO, USA), in line with the manufacturer’s instructions. The E-11 cell line (catalogue no. 01110916) derived from snakehead fish fry (*Channa striata*) was used as a reference for the *Mycoplasma* contamination screening. Positive and negative controls were included in the analysis. The PCR amplicons were separated via 1.5% agarose gel electrophoresis and visualized using a ChemiDoc Imaging System (Bio-Rad, Hercules, CA, USA).

To detect the potential contamination of the snakehead retrovirus in the E-11 and RHTiB cell lines, the total RNA was extracted using GENEzol^®^ Reagent (Geneaid, Taipei, Taiwan), in accordance with the manufacturer’s protocols. The RNA concentration was adjusted to 200 ng/µL with nuclease-free water, and the first-strand cDNA was synthesized using a ReverTra Ace^®^ kit (Toyobo, Osaka, Japan), according to the manufacturer’s recommended protocol. PCR was carried out according to Nishizawa et al. [27] with forward (5′-TGGTACCCATGGATACAGGTACCTCA-3′) and reverse primers (5′-TGTCAGACATGGCCTGTACT) in a T100 thermocycler (Bio-Rad, Hercules, CA, USA). The PCR products were visualized via 1.5% agarose gel electrophoresis and analyzed using the ChemiDoc Imaging System (Bio-Rad, Hercules, CA, USA).

### 2.7. Chromosome Analysis of RHTiB Cells

The chromosome analysis was prepared using the RHTiB cells at passage 25. The cell cultures at 60–70% confluence were treated with 10 μg/mL of KaryoMAX™ Colcemid™ Solution (Invitrogen, ThermoFisher Scientific, Waltham, MA, USA) for 40 min. The cells were subsequently digested, centrifuged, and resuspended in a 0.075 M potassium chloride hypotonic solution for 20 min and fixed in a cold methanol and acetic acid mixture at a 3:1 ratio. The fixed cells were dropped vertically onto glass slides, stained with 5% Giemsa solution for 5 min, and the metaphase chromosomes were observed under a light microscope at 100× magnification using an Olympus VS120 Slide Scanner (Olympus, Tokyo, Japan).

### 2.8. Immunofluorescent Assay to Detect TiLV in RHTiB Cell Line

We used an immunofluorescent assay (IFA) to confirm the susceptibility of the RHTiB cell lines to TiLV. Briefly, 1 × 10^5^ of RHTiB cells were plated on a cell culture chamber slide (SPL Life Sciences, Pocheon-si, Republic of Korea) and allowed to propagate in L-15 medium containing 5% FBS at 25 °C until 80–90% confluency was reached. After washing, the cells were inoculated with the TiLV strain VETKU-TV08 at 0.1 MOI or a control medium for 1 h at 25 °C. The cells were then rinsed and incubated with 2% FBS L-15 medium at 25 °C for an additional 24 h. After fixation with ice-cold 100% methanol for 10 min and permeabilization with 0.3% triton-X 100 for 10 min, a blocking solution (2% bovine serum albumin in PBS) was applied at 25 °C for 30 min. The cells were then probed with rabbit polyclonal antibodies against TiLV [28] in blocking solution at a dilution of 1:100 overnight at 4 °C. The cells were subsequently rinsed three times with PBS and incubated with a secondary antibody (Goat Anti-Rabbit IgG-Alexa Fluor™ 488, Abcam, Carlsbad, CA, USA) in blocking solution at a dilution of 1:500 for 1 h at room temperature. The cellular nuclei were stained with diaminophenylindole (DAPI) at a final concentration of 1 µg/mL (Sigma-Aldrich, St Louis, MO, USA), before visualization under a confocal microscope (Fluoview 3000, Olympus, Tokyo, Japan).

### 2.9. Propagation of TiLV in RHTiB Cell Line under Varying pH and Temperature Conditions

A volume of 500 µL TiLV was inoculated into each well of a 24-well cell culture plate containing confluent RHTiB cells. After incubation for 1 h at 25 °C, the viral suspensions were replaced with L-15 medium containing 2% FBS per well. To investigate the impact of varying pH levels (7, 7.2, 7.4, 7.6, and 7.8) and temperatures (25 °C, 28 °C, and 30 °C), the L-15 medium was added to the 24-well cell culture plates under different pH levels and temperatures. The propagation of TiLV under the different cell culture conditions was confirmed using RT-qPCR on days 1, 5, 7, and 9. The total RNA was extracted from the cell culture supernatant using GENEzol^®^ Reagent (Geneaid, Taipei, Taiwan) and processed for RT-qPCR analysis in line with the published protocol of Yamkasem et al. [29]. The cytopathic effects (CPE) were monitored regularly through an inverted microscope (CKX53, Olympus, Tokyo, Japan) at 40× magnification, with the E-11 cell line as a positive control.

### 2.10. Statistical Analysis

The data on cell growth and virus propagation were analyzed via two-way ANOVA followed by Tukey’s multiple comparisons post hoc tests using the GraphPad Prism software, version 8.3.0 (GraphPad, San Diego, CA, USA). Statistical significance was considered at *p* < 0.05.

## 3. Results

### 3.1. Primary Cell Isolation and Culture

We attempted to isolate primary cell lines from various organs of red hybrid tilapia, including the brain, spleen, heart, fins, and ocular muscles. Interestingly, only the cells derived from the brain exhibited continuous growth without degenerative or morphological changes (Figure 1). The cells from most of the other tissues showed limited proliferation and viability during the 14 days of the primary culture (Appendix A). Consequently, for further experimentation, the brain-derived cells were selected and designated as the RHTiB cell line. When establishing the primary cell culture, the brain tissue explants showed rapid cell proliferation and formed a monolayer of cells (1.2 × 10^6^ cells/mL) within 2–3 weeks of tissue seeding (Figure 1A). The cells subsequently reached 80–90% confluence in successive passages within 7–9 days of subculturing (Figure 1B). The RHTiB cells exhibited a fibroblast-like morphology during their early growth phase, and this remained stable through the subsequent passages. This cell line has been maintained for 1.5 years and has undergone over 50 subcultures.

The viability of the cryopreserved RHTiB cells was assessed after 30 days in storage. Remarkably, approximately 75–80% of the initial RHTiB cell population was successfully revived and demonstrated optimal growth when cultured in L-15 medium supplemented with 20% FBS. The revived cells achieved confluency within 7 days of seeding in a 25 cm^2^ cell culture flask and displayed no observable changes in cell morphology or detachment following cryopreservation and thawing.

### 3.2. Optimization of RHTiB Cell Growth Conditions

The optimal growth conditions for the RHTiB cells were investigated by examining the effects of the FBS concentration, pH, and temperature. We found that the RHTiB cells grew well at 25–30 °C in the L-15 medium supplemented with 20% FBS at pH 7.4. Specifically, we observed a direct link between the FBS concentration and the growth rate. In 20% FBS, the RHTiB cells demonstrated the highest growth rate, with an increase from 3.3 × 10^5^ cells/mL to 1.35 × 10^6^ cells/mL within 7 days, while the cultures with 5–15% FBS exhibited slower growth at 1.8 × 10^5^ cells/mL to 1.05 × 10^6^ cells/mL at 7 days of primary culture (dpc; Figure 2A). The optimal growth of the RHTiB cells occurred at pH 7.4, where the cell numbers increased from 4.1 × 10^5^ cells/mL to 1.33 × 10^6^ cells/mL at 7 dpc with no observed morphological changes or detachment (Figure 2B). Remarkably, despite showing higher live cell counts of 1.58 × 10^6^ cells/mL to 1.96 × 10^6^ cells/mL at 7 dpc, increasing the pH from 7.6 to 8 led to noticeable cell shrinkage and detachment. Finally, on the same day, the number of RHTiB cells cultured at 25 °C, 28 °C, and 30 °C did not differ significantly (*p* > 0.05; Figure 2C). Specifically, the number of RHTiB cells ranged from 5.5 × 10^4^ to 9.15 × 10^4^ cells/mL at day 1, 1.75 × 10^5^ to 1.8 × 10^5^ cells/mL at day 3, 2.7 × 10^5^ to 2.87 × 10^5^ cells/mL at day 5, and 3.01 × 10^5^ to 3.42 × 10^5^ cells/mL at day 7, within these temperatures.

### 3.3. Species Identification of RHTiB Cell Line Using cox1 Gene

We evaluated the origin of the RHTiB cell line through the amplification and sequencing of partial fragments of the *cox1* gene. Our analysis of the *cox1* gene fragments from the RHTiB cells revealed the expected PCR products at 650 bp (Figure 3A). Subsequent BLAST analysis showed significant sequence similarity of 99.22% between the *cox1* nucleotide sequence of the RHTiB and Nile tilapia (GenBank accession number MK355381.1). Furthermore, the genetic distances of the *cox1* DNA sequence between the RHTiB and Nile tilapia were 0.01 (Appendix A). The phylogenetic analysis constructed using the *cox1* nucleotide sequence placed the RHTiB cell line within the *Oreochromis* genus, with marked divergence from both Nile tilapia and other tilapia species (Figure 3B). The nucleotide sequence of the *cox1* gene of RHTiB was deposited at the NCBI GenBank under the accession number OQ351327. Further chromosome analysis revealed that more than 75% of the RHTiB cells had a diploid karyotype during the metaphase, with normal chromosomes at 2n = 44 (Figure 4 and Appendix A).

### 3.4. Detection of Mycoplasma and Snakehead Retrovirus Contamination

*Mycoplasma* and snakehead retrovirus contamination in the RHTiB cell line were confirmed through PCR amplification with the E-11 cell line as a reference. The supernatants collected from both the RHTiB and E-11 cell lines cultured in L-15 media did not exhibit any amplification corresponding to the *Mycoplasma*-specific 259 bp amplicon, as observed in the positive control (Figure 5A). Additionally, for the snakehead retrovirus contamination screening, no amplification was detected in the RHTiB cells. However, a specific amplicon of approximately 730 bp was shown in the E-11 cell line (Figure 5B).

### 3.5. Detection of TiLV Infection in RHTiB Using Immunofluorescence

We evaluated the susceptibility of the RHTiB cell line to TiLV infection using an immunofluorescence technique. The specific rabbit polyclonal antibody against TiLV, labeled with Alexa Fluor™ 488, was incubated to detect the localization of TiLV in infected cells. Although no clear CPE was observed in the infected RHTiB cells (Figure 6A), which was in contrast to the clear CPE in the E-11 cells (Figure 6B), a strong fluorescence signal was observed in the cytoplasm of the RHTiB cells at 24 h post-TiLV inoculation (Figure 6D and Appendix A). No TiLV fluorescence signal was observed in the uninfected RHTiB cells (Figure 6C).

### 3.6. Propagation of TiLV in RHTiB Cells under Different pH and Temperature Conditions

The RHTiB cells were exposed to TiLV under various culture conditions to assess the influence of the pH and temperature on the propagation of the virus. The cultures were maintained at various pH levels (7, 7.2, 7.4, 7.6, and 7.8) and temperatures (25 °C, 28 °C, and 30 °C) for 9 days. The viral copy numbers were quantified using RT-qPCR analysis. On day 7 following TiLV inoculation, a significant (*p* < 0.05) increase in the viral copy numbers was observed at pH 7.6 (10^7.32 ± 0.01^ viral copies/400 ng cDNA) and 7.8 (10^7.30 ± 0.27^ viral copies/400 ng cDNA) compared to pH 7 (10^5.63 ± 0.27^ viral copies/400 ng cDNA), 7.2 (10^5.55 ± 0.02^ viral copies/400 ng cDNA), and 7.4 (10^5.71 ± 0.13^ viral copies/400 ng cDNA). However, on day 9, the highest TiLV copy numbers (10^7.82 ± 0.22^ viral copies/400 ng cDNA) were detected at pH 7, while the lowest levels (10^7.00 ± 0.13^ viral copies/400 ng cDNA) were recorded at pH 7.6 (Figure 7A).

With respect to the influence of the temperature on TiLV propagation, no significant differences (*p* > 0.05) were observed in the TiLV copy numbers in the RHTiB cell line when exposed to temperatures that varied between 25 °C and 30 °C (Figure 7B). Although the highest TiLV copy number was detected on day 9 at 30 °C (10^6.51 ± 0.49^ viral copies/400 ng cDNA), there was no statistically significant difference among the temperatures on this day.

## 4. Discussion

The recent outbreaks of TiLV in several countries and the inclusion of TiLV among the World Organisation for Animal Health’s listed diseases has highlighted the need for TiLV disease surveillance and the development of diagnostic tools. Cell culture is considered the gold standard technique for virus isolation but requires cell lines specific to each virus. In this study, we isolated, characterized, and tested the susceptibility of new cell lines from red hybrid tilapia for TiLV propagation. Red hybrid tilapia, which is commonly cultured in Southeast Asia, is highly susceptible to TiLV. TiLV infection often leads to high morbidity and mortality rates [9]. It is detectable in various organs, including the liver, spleen, head kidney, gills, heart, intestines, and gonads of infected tilapia, with the liver and brain as the major target tissues of this virus [9,30,31]. Establishing a tilapia-susceptible cell line from this species would therefore aid in future studies of the pathogenesis of TiLV [16,19]. Several tilapia cell lines already exist, including TiB [19], On1B [18], and TA-02 [22], from *O. niloticus* and OmB from *O. mossambicus* [32]. In the present article, we report the first brain cell line isolated from red hybrid tilapia, namely the RHTiB cell line, using the tissue explant method. The brain tissue explants demonstrated rapid proliferation during the early growth phase. A monolayer of cells formed within 2–3 weeks of seeding and exhibited a homogeneous population of fibroblast-like cells. Similar findings have been reported in previous studies, which have described a fibroblast dominance in the primary cell cultures derived from tilapia brain tissue [18,19,20,21,22]. Although we successfully isolated primary cells from the brain of red tilapia, our attempts to isolate primary cells from other organs, including the spleen, heart, fins, muscles, liver, and gills, have not yet been successful. The failure to propagate and sustain these primary cells may be influenced by various factors, such as unsuitable culture conditions, the nature of each cell type, the initial cell seeding density, and the choice of growth factors and enzymes used during the subculturing processes [33,34]. Optimizing these factors is important for future efforts aimed at isolating primary cells, which could serve as a tool to investigate the pathobiology of TiLV.

To verify the origin of the RHTiB cells, we amplified and sequenced the mitochondrial *cox1* gene, which is a well-established DNA marker [35]. Additionally, chromosomal analysis confirmed a normal diploid karyotype (2n = 44) consistent with *Oreochromis* spp. The RHTiB cells were also free of *Mycoplasma* contamination and showed promising cryopreservation capabilities, with revival efficiencies reaching 80%, which is comparable to those of other established fish cell lines [18,19,36]. Interestingly, the growth of the RHTiB cells was significantly influenced by the FBS concentration and pH but not the temperature. Similar to other tilapia cell lines [18,22] and cell lines derived from other fish species, including striped catfish (*Pangasianodon hypophthalmus*) [37] and catfish (*Clarias dussumieri*) [38], the RHTiB cells showed maximal growth in the L-15 medium supplemented with 20% FBS. FBS is a complex mixture of nutrients essential for cell growth and thus likely provides an optimal environment for faster proliferation at higher concentrations [39,40]. However, in the interest of long-term cost-effectiveness, we found that 10% FBS was sufficient for RHTiB cell propagation, which is consistent with the findings of a previous study [19]. Similar to primary Nile tilapia cells [19], the optimal pH for RHTiB cell growth was 7.4, which is the physiological pH that supports the optimal activity of most cellular protein and enzyme activities. While tilapia can tolerate a wider pH range between 6 and 8.5 [41], we observed cell shrinkage and detachment at higher pH levels, possibly due to alterations in the cell membrane properties [42]. Nevertheless, additional conditions for cell and virus propagation including different culture media, CO_2_ concentrations, and humidity should be explored for the new cell line.

Similar to the optimal temperatures reported for other tilapia brain cells, the RHTiB cells demonstrated robust growth across a temperature range of 25 °C to 30 °C [19,22]. As a poikilothermic species, tilapia naturally adapts to water temperatures within this range, which is relevant to the observed optimal growth of the RHTiB cells. Similar observations of optimal growth within this temperature range have been reported for various tropical fish species, including tilapia [22,43]. Unlike the E-11 and other tilapia brain-derived cell lines, the RHTiB cells did not exhibit CPE following TiLV inoculation [16,19]. However, both IFA and RT-qPCR revealed RHTiB susceptibility and permissiveness to TiLV propagation. Similarly, in previous studies, IFA was applied to detect TiLV in Nile tilapia-derived TiB cells [16,19] and Mozambique tilapia-derived bulbus arteriosus cells [44], which suggests the potential of these cell lines in further TiLV entry mechanism research.

Further investigations revealed that the pH could impact TiLV propagation in RHTiB cells. We noted a significant increase in TiLV replication within the L-15 medium at a pH ranging from 7 to 7.8, with the highest TiLV concentration observed on day 9 at pH 7. Similarly, despite high variation, the highest TiLV copy number observed in this study was detected on day 9 at 30 °C. While the different tested temperatures did not exhibit significant differences, these findings are consistent with previous reports of optimal TiLV propagation at temperatures between 25 °C and 30 °C [45]. This highlights the potential of the RHTiB cell line as a valuable tool in studying the dynamics of TiLV infection in tilapia. The ability of RHTiB cells to support TiLV replication without displaying CPE makes them a unique resource for the investigation of host cell–virus interactions during prolonged infections and opens up opportunities for future TiLV research.

## 5. Conclusions

We successfully established and characterized the RHTiB cell line, which is the first reported brain cell line derived from red hybrid tilapia. This new cell line holds great potential for various applications, including TiLV detection and vaccine development, and as a tool to study host–virus interactions.

## Figures and Tables

**Figure 1 animals-14-01522-f001:**
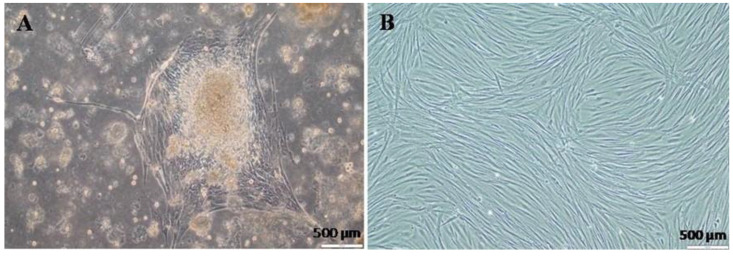
Establishment of the RHTiB cell line derived from the brain tissue of red hybrid tilapia, *Oreochromis* spp. (**A**) The outgrowth of RHTiB cells from a tissue explant on day 3 (40× magnification). (**B**) A confluent monolayer of RHTiB cells (40× magnification).

**Figure 2 animals-14-01522-f002:**
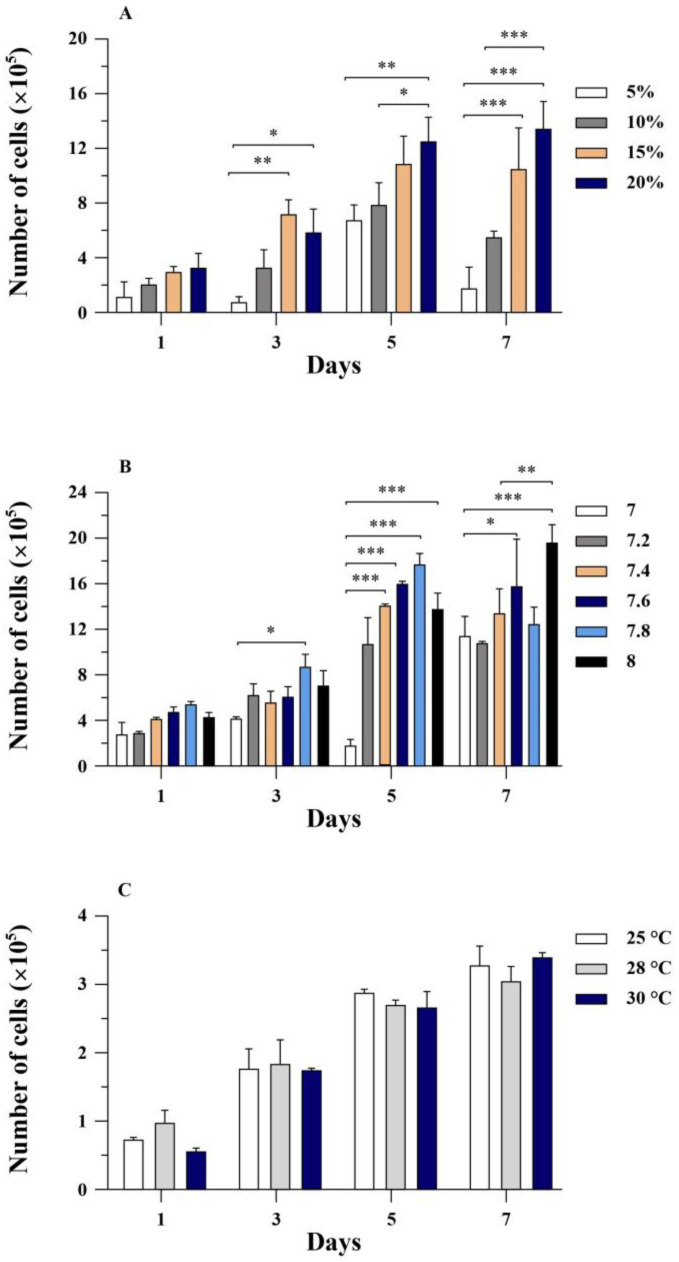
Growth characteristics of RHTiB cells under various culture conditions over a 7-day period. (**A**) Growth of RHTiB cells under different concentrations of fetal bovine serum (FBS; %) in L-15 medium. (**B**) Growth of RHTiB cells under different pH levels in L-15 medium. (**C**) Growth of RHTiB cells at different incubation temperatures (°C) in L-15 medium. The cells were incubated without CO_2_ in an incubator. The mean values with the standard error of the mean (SE) are presented (*n* = 3), and significant differences are denoted as * *p* < 0.05, ** *p* < 0.01, and *** *p* < 0.001.

**Figure 3 animals-14-01522-f003:**
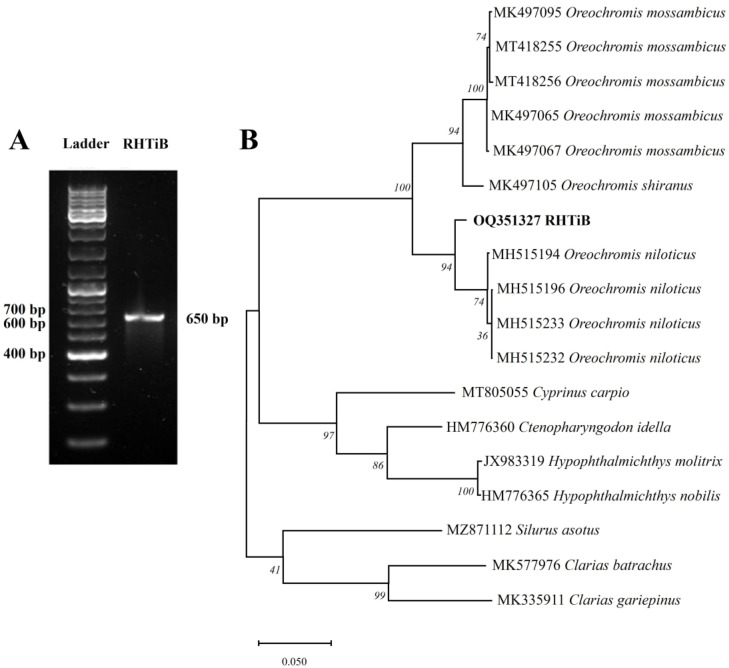
Molecular characterization of RHTiB cell line through amplification of mitochondrial gene. (**A**) Polymerase chain reaction (PCR) amplification of a partial nucleotide sequence of the *cox1* gene from the RHTiB cell line. MW = 100 bp molecular weight marker, lane 1 = 650 bp PCR amplicon from RHTiB cells. (**B**) Phylogenetic analysis based on the *cox1* nucleotide sequence of the RHTiB cell line and a sequence from other fish species using the Molecular Evolutionary Genetics Analysis X tool. The analysis included 18 nucleotide sequences in the final dataset. The GenBank accession number of the reference sequence is provided before each fish name in the phylogenetic tree. The numbers at the nodes show the bootstrap values obtained from 1000 replicates.

**Figure 4 animals-14-01522-f004:**
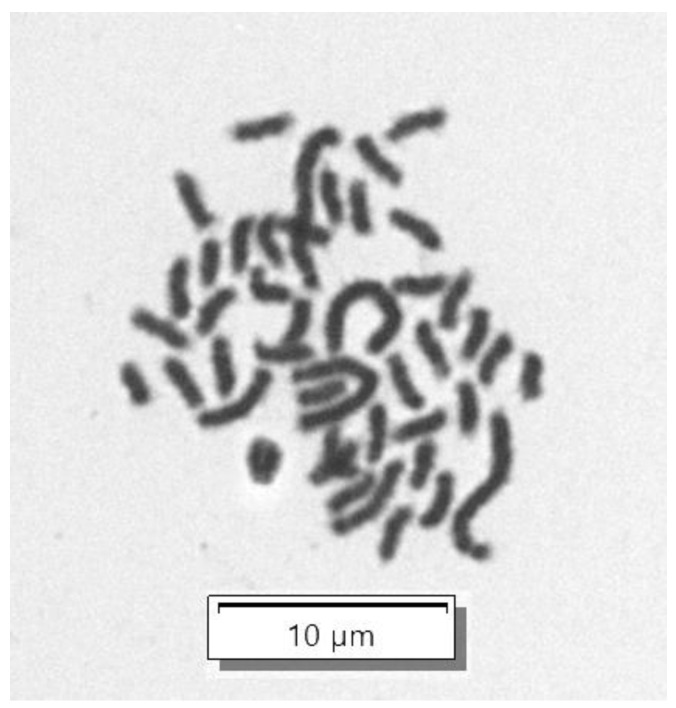
Chromosome analysis of RHTiB cells. Giemsa staining of the RHTiB cells at passage 25 revealed diploid chromosomes (2n = 44) at a magnification of 100×.

**Figure 5 animals-14-01522-f005:**
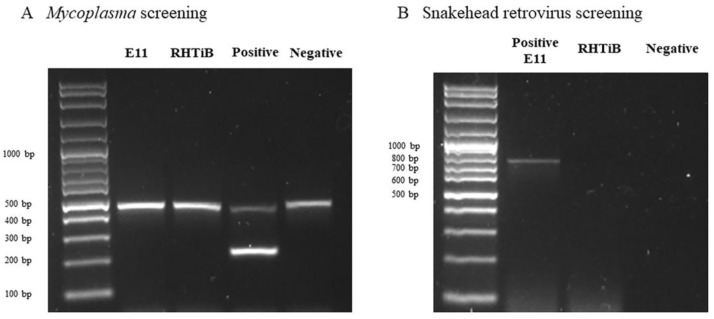
Screening of *Mycoplasma* and the snakehead retrovirus in the E-11 and RHTiB cell lines. (**A**) *Mycoplasma* screening of the E-11 and RHTiB cell lines. A distinct 481 bp band on the agarose gel during *Mycoplasma* screening indicated the presence of internal control DNA, demonstrating the expected performance from the test kit. (**B**) Snakehead retrovirus screening of the E-11 and RHTiB cell lines.

**Figure 6 animals-14-01522-f006:**
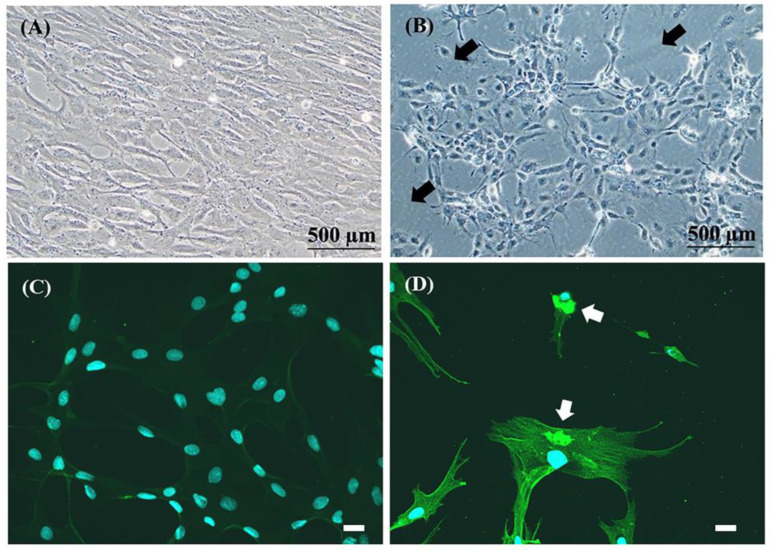
Photomicrographs illustrating the cytopathic effects (CPE) in the cell lines infected with tilapia lake virus (TiLV) and the immunofluorescence staining of the TiLV-infected RHTiB cells. (**A**) The RHTiB cell lines infected with TiLV at 4 days post-infection (dpi), showing no observable CPE formation (40× magnification). (**B**) The E-11 cell line infected with TiLV at 4 dpi, displaying CPE in the form of cell lysis (black arrow), cell rounding, clumping, and the formation of syncytial cells (40× magnification). (**C**) Uninfected RHTiB cells showing diaminophenylindole (DAPI) staining of the nuclei of the normal cells. (**D**) Discrete green fluorescence signals (white arrows) in the cytoplasm of infected cells after 24 h of virus inoculation. The blue color indicates the DAPI staining of the nucleus. The scale bar represents 20 µm.

**Figure 7 animals-14-01522-f007:**
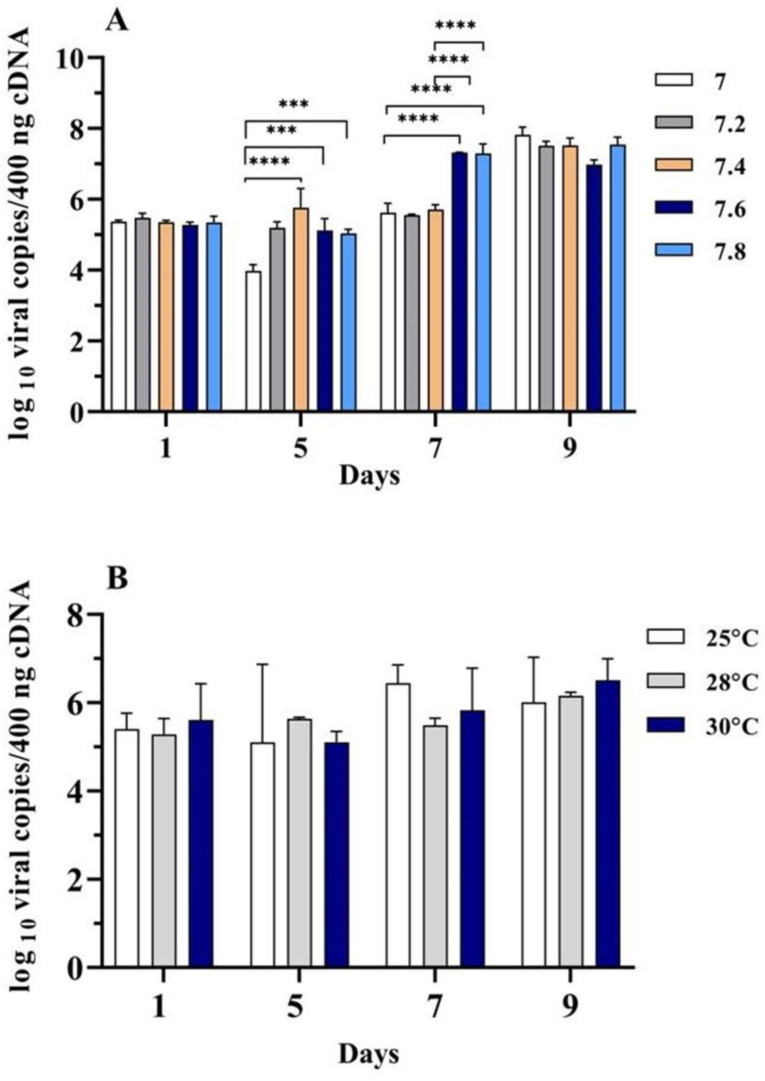
Quantification of the tilapia lake virus (TiLV) copy number (log_10_ viral copies/400 ng of cDNA) in RHTiB cell lines cultured under different conditions. (**A**) Viral copies in the RHTiB cell line at five pH levels. (**B**) Viral copies in the RHTiB cell line grown at three different incubation temperatures. The values indicate significant differences (*** *p* < 0.001 and **** *p* < 0.0001) and are presented as the means ± standard error of the mean (SE; *n* = 3).

## Data Availability

The data presented in this study are available on request from the corresponding author. The data are not publicly available due to the anonymity granted to all participating parties.

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
