# Peer review of "A New Cell Line from the Brain of Red Hybrid Tilapia (Oreochromis spp.) for Tilapia Lake Virus Propagation"

_animals, 2024, doi:10.3390/ani14111522_

Round 1
Reviewer 1 Report
Comments and Suggestions for Authors
The authors established a new cell line derived from red hybrid tilapia brain tissue, RHTiB, and used this cell line to propagate the virus. This work is a piece of nice work and would facilitate investigating on fish virus in the future. However, in this manuscript, there some points needed to be verified before it is accepted as a publication in the Journal.
1. The authors should show all the cells they observed for the chromosome counting and the number of chromosome in the cells, but not just show 22 pairs of chromosome in one cell (n=44).
2. The authors should determine the efficiency of virus effecting cells and provide a picture with more cells inffected with virus since the cells would be used to propagate the virus.
Reviewer 2 Report
Comments and Suggestions for Authors
Animals:
New cell line from the brain of red hybrid tilapia (Oreochromis spp.) for tilapia lake virus propagation
Mohamad et al.
This manuscript describes attempts to develop cell lines from red hybrid tilapia tissues and potential for entry and replication of Tilapia Lake Virus. Primary cultures of most tissues did not result in useful continued cell cultures, but brain tissue explants did result in useful confluent cell cultures that were tested for optimal incubation temperatures, pH of L-15 media, level of FBS in media, and virus replication. My main concerns are that in the temperature optimization step, the virus copy levels did not go up appreciably with time. The pH study did have increased virus copy levels with time, but the trend for optimal pH was not clear since sometimes pH 7.0 was best but other days other pH was higher. Seems it would be more obvious and consistent from day to day. The results were clear that more replicates were needed to lower the standard error to allow more significance, especially for the temperature study. It is okay to report what you did and the results obtained though may not be as desirable as hoped. Other conditions or enhancements to methods may be required for higher propagation of the virus. Seems like this virus can get into the cells and you showed that, but not replicating very well under these conditions.
This manuscript is very well written and highly detailed. I really enjoyed reading this paper. I will provide most of my comments by listing the line number first.
Line 12 – “optimal” growth but didn’t test other media besides L-15, 20% FBS was shown to be best (albeit more costly), and incubation temperatures were basically all the same for growth. If you had included lower or higher temps, you might have shown that 25 to 30 C were significantly better.
Line 23 – optimization included many of conditions I would expect but didn’t show other media types besides L-15. I know L-15 may be better, but not proven here.
Line 30 – maximum viral load at 9 days, but really didn’t test beyond that number of days to show it really is the high point.
Line 57 – said something about needing specific cell line for host trying to detect virus, but this seems to this reviewer not to be true. The virus may have very narrow host preference, but others may be wide like VHSV detected in dozens of hosts.
Line 74, 86, 103 – streptomycin units were not stated the same way, perhaps line 74 one is wrong. Also, shouldn’t it be in mg/mL not just mg?
Line 87 & 97 – 2000 RPM but better if in RCF instead.
Line 99 – 90% FBS seems quite high to me, just checking that no other media was present in freezing solution.
Line 112 – Your normal L-15 media was pH 7.4, yet you state you adjusted the pH to 7.0 to 8.0 by using only 1N HCl. Seems like perhaps you started at pH 8 and added HCl or used HCl and NaOH for adjustments.
Line 184 – should include the volume of inocula for these wells.
Line 240 – High cell numbers in limited space in wells can lead to shrinkage and rounding of cells, but you are right that you don’t want them to detach.
Line 241 – No apparent preference for 25, 28, 30, but had high variation.
Line 250 – Don’t see any in this figure with ***?
Line 264 – Figure 3 description surrounds the actual figure, seems odd to me.
Line 264 – How did you select which cox1 sequences to include? More species must be available, but were these the closest in blastn search or how to decide on other tilapia and non-tilapia species?
Line 282 – explain the 500 bp band, is it a housekeeping gene, but then why was it in the negative control?
Line 306 – “Dispute” maybe wrong wording, perhaps “Discrete”?
Line 315 – pH had different effect on day 7 and day 9? Very odd switch for pH 7.0 better on day 9, but pH 7.6 and 7.8 clearly better on day 7. Why in only 2 days? Is day 1 mostly from the initial inoculum attached to cells, very consistent on day 1, then some go lower on next sampled day.
Line 327 – Figure 7B did not increase over time at these 3 temps, not compared to the increase in Figure 7A at only one temp with varied pH level. Why higher variation occurred for 7B compared to 7A?
Line 379 – cell shrinkage and shape alterations can occur due to space limitation in wells. Observed this with other fish fibroblast-like cells.
Line 389 – Didn’t see much, if any, propagation in the temp part of study!
Line 392 – Misleading, since pH 7.0 was not good on day 7, but best on day 9. Others jumped around quite a bit for pH in short timeframes. What about beyond day 9, did it go down or continue to get higher? Not shown.
Line 393 – temperature maximum also was questionable due to high standard error values. Can’t say 30 C on day 9 was best!
Round 2
Reviewer 1 Report
Comments and Suggestions for Authors
This manuscript is acceptable now since the authors have responded to all questions as required.